# Use of Algae in Aquaculture: A Review

Srirengaraj Vijayaram [1], Einar Ringø [2,*], Hamed Ghafarifarsani [3], Seyed Hossein Hoseinifar [4], Saman Ahani [5] and Chi-Chung Chou [1,*]

1. Department of Veterinary Medicine, College of Veterinary Medicine, National Chung-Hsing University, 145 Xingda Rd., Taichung 40227, Taiwan; vijayarambiotech@gmail.com
2. Norwegian College of Fishery Science, Faculty of Bioscience, Fisheries, and Economics, UiT the Arctic University of Norway, 9037 Tromsø, Norway
3. Department of Fisheries, Faculty of Natural Resources, Urmia University, Urmia 57561-51818, Iran; hamed_ghafari@alumni.ut.ac.ir
4. Department of Fisheries, Faculty of Fisheries and Environmental Sciences, Gorgan University of Agricultural Sciences and Natural Resources, Gorgan 49189-43464, Iran; hossein.hoseinifar@gmail.com
5. School of Veterinary Medicine, Islamic Azad University Karaj Branch, Karaj 31499-68111, Iran; samanahani@gmail.com
* Correspondence: einar.ringo@uit.no (E.R.); ccchou@nchu.edu.tw (C.-C.C.)

**Abstract:** The utilization of algae in aquaculture is environmentally friendly, safe, and cost-effective and can effectively substitute for fish meal and fish oil in aquatic feeds. Incorporating algae as dietary supplements leads to significant enhancements in aquatic animals' health and also improves the aquatic ecosystem. Algae are rich sources of nutrients and serve as the foundational food source in the aquatic food chain. Currently, 40 different algae species are employed in aquaculture. Furthermore, algae contributes to elevating the overall quality of aquatic feed products. Aquaculture stands as the most vital food production sector globally; however, challenges such as infection outbreaks and aquatic environmental pollution pose significant threats to the sustainable growth of this industry. An alternative strategy for mitigating environmental issues and improving aquatic production involves the utilization of algae. The novelty in the applications of algae in aquaculture stems from their multifaceted roles and benefits, such as their capacity to improve water quality, serve as nutrient-rich feed supplements, and enhance the overall health and productivity of aquatic species. These versatile applications of algae represent a fresh and innovative approach to sustainable aquaculture practices. This review furnishes insights into the use of algae, algae extracts, or components derived from algae to enhance water quality. Additionally, it covers the utilization of algae-based feed supplements, boosting of the immune system, enhanced growth performance, and disease resistance in aquatic animals.

**Keywords:** algae; water treatment; dietary additives; aquatic animal; aquaculture

**Key Contribution:** Algae are naturally rich nutrient sources; they are the primary food producer in the food chain for aquatic animal life, and the algae cultivation method is eco-friendly, non-toxic, and cost-effective. They have numerous beneficial properties, including immunostimulant, antioxidant, anti-inflammatory, and antimicrobial activity in aquatic animals, and microalgae convert atmospheric carbon into high-nutrient products. Algae improve the circular bioeconomy in the aquaculture industry, and the algae treatment process is a successful method for different types of wastewater from municipal, industrial, agro-industrial, and livestock sources. Both microalgae and macroalgae as well as algae-producing components like carbohydrates, lipids, and proteins are beneficial substances in aquatic feeds and improve the quality of aqua feed, aquatic animal health, and the aquatic environment.

## 1. Introduction

The aquaculture sector plays a crucial role in global food production and holds significant economic importance for many countries. Both finfish farming and crustacean production contribute substantially to the global food supply chain, providing essential protein sources and supporting livelihoods. As the global population continues to grow, the aquaculture sector is expected to play an increasingly important role in meeting the demand for nutritious and sustainable protein sources. Sustainable and responsible aquaculture practices are critical for the long-term health of the industry and the environment [1]. Aquatic feeds typically play a crucial role in augmenting aquatic production, with a substantial portion of fish oil (FO) and fish meal (FM) being used as essential feed ingredients to meet the dietary requirements of farmed aquatic species. Consequently, there is a growing imperative to re-evaluate the sustainability of fish farming regarding the consumption of FM and FO [2]. The utilization of algal technology in aquaculture offers a range of benefits and represents an alternative approach to enhancing various biological functions. Algae, including microalgae and macroalgae, can serve multiple purposes in aquaculture systems, contributing to improved health and environmental conditions. The application of algal technology in aquaculture aligns with the goals of sustainability, environmental stewardship, and improved efficiency within the industry. Research and ongoing developments continue to explore innovative ways to integrate algal technologies for the benefit of both aquaculture production and the surrounding ecosystems [3]. Algae, including brown algae (Phaeophyceae), green algae (Chlorophyceae), red algae (Rhodophyceae), and diatoms (Bacillariophyceae), can be broadly categorized into two primary types: microalgae and macroalgae. The distinction between microalgae and macroalgae is primarily based on size, with microalgae being microscopic and macroalgae being visible without the aid of a microscope. Both microalgae and macroalgae play important roles in aquatic ecosystems and have various applications in industries, ranging from food and agriculture to biotechnology and environmental management [4]. The distinctive or combined effects of various algae species contribute to the consistency of protein, lipids, and essential trace minerals in aquaculture. Some of the commonly used algae species include *Tetraselmis* sp., *Pavlova* sp., *Chlorella* sp., *Isochrysis* sp., *Chaetoceros* sp., *Phaeodactylum* sp., *Skeletonema* sp., *Thalassiosira* sp., and *Nannochloropsis* sp. [5]. Microalgae, as a diverse group of photosynthetic microorganisms, have played crucial roles (photosynthetic evolution, primary producers, formation of planktonic ecosystems, biodiversity, carbon fixation, nitrogen fixation, symbiotic relationships, biogeochemical cycling, and adaptation to diverse environments) in the evolutionary history of life on Earth. While their exact evolutionary timeline is complex and not fully understood, microalgae have significantly contributed to the development and maintenance of ecosystems. Understanding the evolutionary roles of microalgae is crucial not only for elucidating the history of life on Earth but also for recognizing their ongoing ecological importance in contemporary ecosystems and their potential applications in various industries, such as bioenergy, food, and environmental management. Microalgae are indeed recognized as an essential and indispensable source of nutrition for fish fry and larval shrimp in aquaculture. In the context of aquaculture, particularly in hatcheries where controlled conditions are maintained for the early stages of fish and shrimp development, microalgae are considered a key component in ensuring successful and healthy aquaculture production. [6]. Sivaramakrishnan et al. [7] reported that four green algae, *Acetabularia acetabulum*, *Enteromorpha*, *Halimeda macroloba*, and *Halimeda tuna*, exhibit significant potential in terms of antioxidant capabilities, immune stimulation, and medicinal applications for aquatic animals. Algae represent vital natural food sources for aquatic animals, and various algae species, such as *Leptolyngbya valderiana*, *L. tenuis*, *Arthrospira maxima*, *Navicula minima*, *Nostoc ellipsosporum*, *Cytoseira*, *Ulva*, *Pavlova*, *Chaetoceros*, *Porphyridium*, *Chlorella*, *Palmaria*, *Gracilaria*, and *Isochrysis*, are promising feed components, enhancing the growth performance and immunity of Nile tilapia (*Oreochromis niloticus*), rainbow trout (*Oncorhynchus mykiss*), European seabass (*Dicentrarchus labrax*), golden gourami (*Trichopodus trichopterus*), cichlid (*Cichlidae*) fish, wag swordtail (*Xiphophorus hellerii*), orange molly pink zebra, tetras

(*Paracheirodon axelrodi*), and prawns (*Dendrobranchiata*). Therefore, algae biomass serves as a cost-effective, safe, and environmentally friendly feed ingredient [8], and administrations of microalgae directly to the rearing water or indirectly by mixing them into the feed has become a crucial practice for enhancing aquatic animal nutrition [9–11]. Both micro- and macroalgae are advantageous components in aqua feeds, as they contribute to improving feed quality, aquatic animal health, and the aquatic environment [12].

Microalgal components offer not only beneficial effects for aquatic animals but also significant economic and ecological advantages in aquaculture [13]. Microalgal biomass contains a variety of biochemical components, such as proteins, lipids, carbohydrates, natural antioxidants, and bioactive products, and these diverse components are harnessed to develop sustainable aquaculture [14]. Microalgae supplements have been shown to enhance the growth performance of aquatic animals, and they synthesize a wide array of beneficial biological elements, including amino acids, vitamins, antioxidants, and pigments, which are advantageous for the health and growth of aquatic animals [15,16]. Compounds generated by macroalgae, particularly polysaccharides, can up-regulate the immune system and enhance protection against *Aeromonas hydrophila*, *Enterobacter* sp., *Pseudomonas aeruginosa*, *Streptococcus* sp., *Escherichia coli*, *Vibrio parahaemolyticus*, *Vibrio alginolyticus*, *Vibrio cholerae*, *Yersinia enterocolitica*, and *Proteus* sp. [17]. Astaxanthin synthesized by algae enhances antioxidant activity in rainbow trout and offers beneficial effects for human health [18–20]. Furthermore, the inclusion of algae and their various biomolecules, proteins, peptides, amino acids, fatty acids, sterols, polysaccharides, oligosaccharides, phenolic compounds, photosynthetic pigments, vitamins, and minerals collectively contribute significantly to the growth performance, antioxidant activity, and antimicrobial activity, serving as immune-stimulating agents in both aquatic animals and humans [21,22].

Algae are reported to exhibit antibacterial properties against a wide range of bacteria, encompassing both Gram-positive and Gram-negative strains [23,24]. Various algal species synthesize antimicrobial compounds such as short-chain and long-chain fatty acids, which are effective against pathogenic bacteria, including *Vibrio* species, *Aeromonas* species, and *Pseudomonas* species, as well as viruses, including *influenza* B, *mumps* viruses, herpes simplex virus type 1 (HSV-1), and even the human immunodeficiency virus type 1 (HIV-1) [25]. Algae-synthesized polyunsaturated fatty acids (PUFAs), such as eicosapentaenoic acid (EPA) and sterols, have microbicidal effects against several bacteria, including Gram-positive strains like *Staphylococcus aureus* and *Streptococcus pyogenes*, as well as Gram-negative genera such as *Aeromonas*, *Pseudomonas*, and *Vibrio* [26–30].

Microalgae cultivation is used in aquaculture wastewater treatment [31,32], as it can improve water quality [33]. Five marine microalgae strains, *Porphyridium aerugineum*, *Nannochloropsis granulate*, *Tetraselmis chuii*, *Botryococcus braunii*, and *Phaeodactylum tricornutum*, can produce varying levels of essential minerals, including calcium (ranging from 0.26% to 2.99%), phosphorus (0.73% to 1.46%), magnesium (0.26% to 0.71%), potassium (0.67% to 2.39%), sodium (0.81% to 2.66%), and sulfur (0.41% to 1.38%). These microalgae strains produce a total of 26 different chemical compounds, and their biomass plays an important role in carbon dioxide ($CO_2$) conversion, nutrient recycling, land cultivation, and the remediation of wastewater, demonstrating their significance in environmental recycling [34–37]. The use of algae grown on wastewater for aquafeed applications involves several regulatory aspects (wastewater quality standards, nutrient content and composition, microbial safety, toxic substances, harmful algal blooms, feed safety regulations, environmental impact assessment, labeling and documentation, approval and certification, traceability and record keeping, and public health and consumer safety), which need to be considered to ensure both the safety of the aquafeed and the protection of the environment. Regulatory frameworks can vary by country, and specific guidelines may be provided by relevant authorities. Below are some general regulatory considerations. It is essential for individuals or companies engaged in the production of aquafeed using algae grown on wastewater to familiarize themselves with the specific regulations applicable in their region and to work closely with relevant regulatory authorities to ensure compliance. Additionally, seeking

certification from reputable organizations may enhance the credibility and market acceptance of such products [37,38]. The present review aims to provide an overview of recent findings and applications related to the use of both micro- and macroalgae in aquaculture, including the use of macro- and microalgal extracts as functional feed additives as reported by Monteiro et al. [39] and the introduction of innovative approaches that can contribute to the development of sustainable aquaculture.

## 2. Cultivation Process

The cultivation of algae is the process of growing algae for various purposes, including food production, biofuel, bioremediation, and pharmaceuticals. Algae can be cultivated in open ponds, closed bioreactors, or photobioreactors [40]. The cultivation process involves the following elements: strain selection, growth medium, cultivation system, light and temperature, aeration and mixing, harvesting, and processing. Algae cultivation has gained attention as a sustainable and environmentally friendly way to produce a wide range of products with industrial applications, including biofuel production, food and feed production, wastewater treatment, and pharmaceuticals. Additionally, algae cultivation can contribute to carbon capture and reduce greenhouse gas emissions when used for biofuel production, and as a sustainable and versatile practice, algae cultivation has garnered interest in research and industry for its potential to contribute to a range of sectors. Advances in cultivation technologies and the exploration of novel algal strains continue to shape the field, making algae cultivation a promising solution for addressing global challenges related to food security, environmental sustainability, and renewable energy production [41–44]. In addition, it is worth mentioning that Silkina et al. [45] displayed that far-red light adaptation improved phycocyanin, a pigmented protein complex, and xanthophyll concentrations compared to white-light conditions in the thermophilic cyanobacteria Chlorogloeopsis fritschii.

## 3. Wastewater Management in Aquaculture

Aquaculture produces substantial volumes of wastewater, which contains nutrients, organic substances, and living organisms. The wastewater contents can vary, influenced by factors such as the species cultivated, the type of feed utilized, and the methods employed for water management [42,43,45,46]. The use of algal cultivation for the enhancement of water quality, especially in reducing Chemical Oxygen Demand (COD), involves considerations of scale application efficiency (system design, nutrient loading rates, algal species selection, harvesting, and recirculation system), commercial exploitation (nutrient recovery, aquaculture and agriculture, research and development, public and private partnerships), and regulatory aspects (water quality standards, environmental permits, biosafety and health regulations, monitoring and reporting, land use regulations, and product safety). By addressing the scale application efficiency, commercial exploitation, and regulatory aspects, the integration of algal cultivation for water quality improvement can be more effectively implemented and sustained over time. Collaboration with regulatory bodies and adherence to standards are crucial for the successful and responsible deployment of algae-based water treatment solutions [32]. Microalgae cultivation has the potential to improve aquaculture wastewater, as it can affect the excess amount of nutrients such as ammonia, nitrate, phosphate, and chemical oxygen demand (COD) in order to enhance water quality. This removal not only benefits water quality but also mitigates the environmental impact of nutrient-rich effluents when discharged. The harvested microalgae biomass can serve as a valuable feed source for aquaculture species, or it can be utilized for various other purposes, making it a cost-effective and environmentally friendly solution for both wastewater treatment and resource utilization in the aquaculture industry [32,47]. Effective management of aquaculture wastewater is imperative to mitigate the detrimental effects on aquatic ecosystems. Employing efficient waste capture, water treatment, and recycling systems can significantly decrease the discharge of pollutants. In addition, regulatory measures and adherence to regulation are crucial for ensuring the long-term environmental

sustainability of aquaculture operations [38,48,49]. Algal biomasses serve a dual purpose in enhancing the nutritional value of aquatic feed supplements, contributing to both environmental sustainability and the efficient utilization of aquaculture resources [50]. The use of algal technology for removing excess nutrients and dissolved carbon from wastewater is not a recent development; it was first suggested in the 1950s and has since evolved and gained prominence as an environmentally friendly and effective method for wastewater treatment and resource re-utilization [51–54]. Microalgae removes biological matter from wastewater through processes like consumption, adsorption, and biodegradation through microalgae's production of extracellular polymers, which can detoxify or modify pollutants [55–58]. The microalgae *Chlorella vulgaris* has shown impressive efficiency in removing total nitrogen (86.1%) and phosphorus (82.7%) from white-leg shrimp (*Penaeus vannamei*) farming wastewater [59]. Furthermore, *C. vulgaris* demonstrated superior growth in aquaculture wastewater when bicarbonate was used as the carbon source, underscoring its potential for efficient nutrient removal [60]. The microalgae *Scenedesmus* sp. is effective in removing pollutants from wet market wastewater, removing 74.8% of total nitrogen (TN) and 92.2% of total phosphorus (TP) [61]. The microalgae *Spirulina* sp. is also utilized for treating aquaculture wastewater and reducing environmental impact [48].

Microalgae biomass and its components, including astaxanthin, phycocyanin, carbohydrates, proteins, lipids, and PUFAs, are effectively employed to remove eutrophication pollutants from wastewater. The biomass of microalgae species like *Chlamydomonas reinhardtii, Monoraphidium griffithii,* and *Selenastrum* sp. can be cultivated and recirculated for use in treating and improving aquaculture wastewater [62]. The use of the microalgae *Chlorella sorokiniana* in an aquaculture estuary revealed significant effectiveness in nutrient removal, as it showed removal rates of 75.6% for ammonium, 84.5% for nitrates, 73.3% for phosphates, and 71.9% for chemical oxygen demand (COD). [46]. The cultivation of *Chlorella vulgaris* from Nile tilapia effluents showed impressive results in terms of reducing nutrient levels and bioremediation efficiency. Microalgae biomass produces an average nutrient level of 172.91 mg/day for carbohydrates, 141.57 mg/day for proteins, and 150.19 mg/day for lipids. Furthermore, it achieved exceptional bioremediation efficiency, removing 99.8% of nitrate and 99.7% of phosphate from the tilapia effluents. These findings highlight the potential use of *C. vulgaris* in water quality control [63]. Cultivation of *Spirulina platensis* in aquaculture wastewater also demonstrated a similar beneficial effect [64,65]. The use of different algal species in combination has also been revealed to be effective in bioremediation and biomass production [66–68]. Cultivating microalgae in an aquaponic system proves beneficial to both fish and plants. Integrated fish and vegetable farming supports sustainable aquaculture and hydroponic plant growth in a mutually beneficial ecosystem [69–71] (Table 1).

**Table 1.** Biochemical composition and removal nutrient efficiency (%) of algae cultivated in wastewater.

| Algae | Wastewater Source | Biochemical Composition | | | | | Nutrient Removal Efficiency | | | References |
| | | Protein (%) | Lipid (%) | Carbohydrate (%) | COD% | Nitrate (%) | Nitrite (%) | Phosphate (%) | Ammonia (%) | |
|---|---|---|---|---|---|---|---|---|---|---|
| *Chlorella vulgaris* | Aquaculture waste water | 47.5 | 9.1 | 19.1 | 46–76 | 100 | 100 | 63.1–92.2 | 23.4 | [58] |
| *Chlorella* sp. | Aquaculture wase water | 55.28 | 10 | 24.77 | 46–76 | 62 | 82 | 63.1–92.2 | 84–96 | [72] |
| *Chlorella vulgaris* and *Tetradesmus obliquus* | Aquaculture wastewater | 57 | 8 | 16.8 | 94 | 100 | - | - | 99 | [62] |
| *Gracilaria birdiae* | Aquaculture wase water | 12.94 | 4.82 | 58.12 | 78 | ~100 | | 93.5 | 34 | [73] |
| *Spirulina platensis* | Nile tilapia effluent | 56 | 13 | 25 | 87.2 | 97.5 | 95 | 98 | 98 | [63] |
| *Scenedesmus* sp. | Tilapia | 19.5 | 30.9 | 35.15 | 80 | 77.7 | 73.8 | ~100 | 88.7 | [59] |

Abbreviation: COD—Chemical Oxygen Demand.

## 4. Applications of Algae in Aquaculture

### 4.1. Replacements of Fish Meal and Fish Oil

Algae, particularly microalgae, are essential natural nutritional sources in aquatic ecosystems and play an important role in the food chain of aquatic animals. The beneficial effects include improved growth performance, modulation of gut microbiota, and enhanced disease resistance, making them valuable for the aquaculture industry. Furthermore, the substitutions of FM and FO with microalgae contribute to the conservation of marine ecosystems [74–76]. Dietary additives of algae have demonstrated effectiveness as immunomodulators by enhancing survival rates of olive flounder (*Paralichthys olivaceus*) exposed to *Edwardsiella tarda* [77]. Micro- and macroalgae used as a feed supplement have demonstrated positive effects on the green water culture production of various planktivorous species, including Nile tilapia, rohu (*Labeo rohita*), bighead carp (*Hypophthalmichthys nobilis*), catla (*Catla catla*), and shrimp. Microalgae such as Arthrospira (*Spirulina*), *Chlorella*, *Dunaliella*, *Haematococcus*, *Isochrysis*, *Nannochloropsis*, *Pavlova*, and *Schizochytrium* have established themselves as essential nutritional sources, with benefits for aquatic animals [5,18].

#### 4.1.1. Microalgae

In a previous review, Duerr et al. [78] discussed the use of the microalgae *Chaetoceros*, *Thalassiosira*, *Tetraselmis*, *Isochrysis*, and *Nannochloropsis* as aquaculture feeds, through *Artemia*, rotifers, or *Daphnia*. The dietary inclusion of microalgae supplements represents an innovative and sustainable approach in aquaculture and has garnered increasing attention as an alternative to FM and FO in commercial aquatic feed, enhancing the nutritional value of the feed and promoting environmental sustainability in aquaculture [79–81], which moves the industry closer to achieving sustainable aquaculture [82]. Microalgae species such as *Chlorella*, *Dunaliella*, *Nannochloropsis*, *Spirulina*, and *Scenedesmus* are commonly used in aquaculture as feed supplements to improve the health and growth performance of aquatic animals [83]. The combination of algae extracts and organic acids has the potential to partially or completely replace FM and FO in aquafeed [84]. Biomass from species like *Dunaliella* spp. can serve as a valuable ingredient in aquaculture feeds, particularly for Nile tilapia, and its incorporation can partially replace FM, leading to improved growth and nutrient intake in the cultured species. This practice supports the development of a more sustainable and environmentally friendly aquaculture industry by reducing the dependency on traditional protein sources and promoting resource efficiency [85]. In a subsequent study, it was reported that the incorporation (10%) of *Dunaliella* sp. in Nile tilapia diets is safe and economically beneficial, as the additive improved fish health and growth performance and reduced the supplementation levels of FO and FM in the diets. This aligns with sustainability goals in aquaculture, emphasizing the responsible use of resources and reducing the reliance on traditional feed ingredients [5].

#### 4.1.2. Macroalgae

The incorporation of macroalgae as a feed additive in an 8-week feeding trial at a dosage of 1000 mg led to significant improvements in growth, antioxidant capacity, and pigmentation in electric yellow cichlid (*Labidochromis caeruleus*) [78]. The combination of dietary supplements containing macroalgae species such as *Ulva* spp., *Gracilaria* spp., and *Fucus* spp. at concentrations ranging from 2.5% to 7.5% yielded several positive effects in European sea bass (*Dicentrarchus labrax*). One of the significant outcomes is the potential to replace traditional feed ingredients FM and FO [86]. The incorporation of algae supplements not only enhances the health and performance of aquaculture species but also contributes to making aquaculture practices more sustainable by reducing the industry's reliance on marine resources. This aligns with the growing emphasis on environmentally responsible aquaculture [84,85,87]. In a previous study with Nile tilapia, the dietary effect of ulvan extract from the green algae *Ulva clathrata* revealed that administration increased hematocrit, phagocytic activity, and white blood cells. However, no significant effect was displayed for growth performance, hemoglobin, red blood cell count, total serum protein,

albumin, and globulin in the dietary levels as a result of the ulvan extract [87,88] (Figure 1, Table 2).

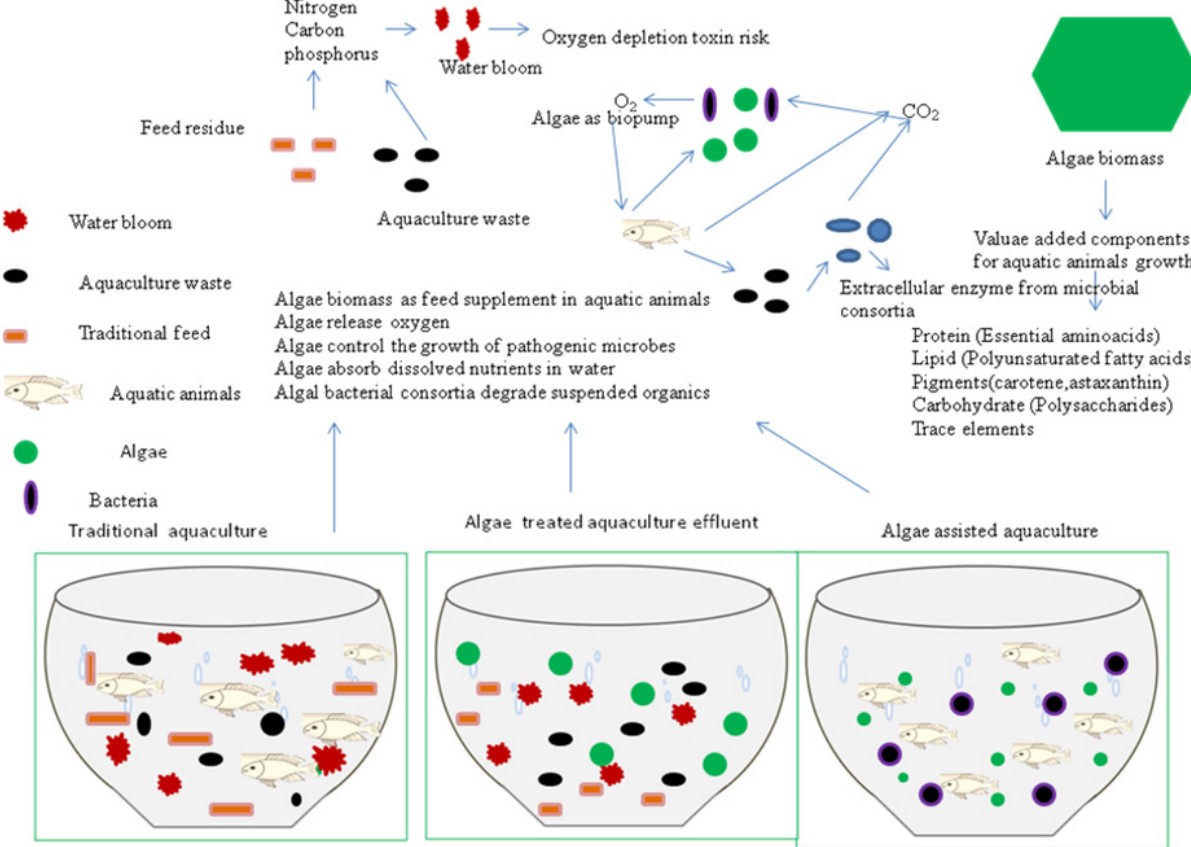

**Figure 1.** Applications of algae in aquaculture. Algae play diverse and significant roles in aquaculture, contributing to water quality management (nutrient removal, BOD, $CO_2$ sequestration, and bioremediation) and acting as valuable feed supplements (natural nutrition source of omega-3 fatty acids, pigments, and carotenoids) and providing enhanced growth and reproduction, artemia enrichment, biofloc technology, seedling production, and sustainable practices for aquatic animals. The integration of algae in aquaculture demonstrates its versatility in addressing multiple challenges, from water quality management to providing essential nutrition for aquatic animals. Research and ongoing developments continue to explore ways to optimize the use of algae for sustainable and efficient aquaculture production.

**Table 2.** Algae is used as a dietary feed supplement in aquaculture and its effects on aquatic animals.

| Algae | Algae (Family) | Mode of Algae in Aqua Feed | Aquatic Organisms (Scientific Name) | Aquatic Organisms (Common Name) | Dietary Inclusion Level (g/kg or %) | Replaced Ingredients | Beneficial Effects | References |
|---|---|---|---|---|---|---|---|---|
| *Arthrospira platensis* | Microcoleaceae | Whole | *Macrobrachium rosenbergii* | Giant river prawn | 50% | FM | Increased growth performance and feed conversion efficiency | [89] |
| *Arthrospira platensis* | Microcoleaceae | Whole | *Oreochromis niloticus* | Nile tilapia | 10% | FM | Improve growth rate and digestive enzyme activity | [90] |
| *Chlorella vulgaris* | Chlorellaceae | Whole | *Carassius auratus gibelio* | Carp | 0.4–2.0% | FM | Enhanced immunoglobulin (Ig) M and D, interleukin-22 (IL)-22, and chemokine (C-C motif) ligand 5 (CCL-5) | [91] |
| *Chlorella vulgaris* | Chlorellaceae | Whole | *M. rosenbergii* | Giant river prawn | 4–8% | FM | Enhanced growth rate, immune system, and disease resistance against pathogens | [92] |
| *Dunaliella salina* | Dunaliellaceae | Whole | *Penaeus monodon* | Black tiger shrimp | 10% | _ | Improved growth rate | [93] |
| *Gracilaria fisheri* | Gracilariaceae | | *P. monodon* | Giant tiger prawn | 100–200 µg/ml | FM - | Enhanced immune system and increased resistance against white spot syndrome virus | [94] |
| *Gracilaria arcuata* | Gracilariaceae | | *O. niloticus* | Nile tilapia | 20% | FM | Enhanced growth performance | [95] |
| *Nannochloropsis oculata* | Monodopsidaceae and Thraustochytriaceae | LEA | *O. niloticus* | Nile tilapia | 33–100% | PR | Enhanced growth rate and nutritional quality of farmed fish | [96] |
| *Pyropia spheroplasts* | Bangiaceae | Whole | *Apostichopus japonicus* | Japanese sea cucumber | 50 g/kg | - | Improved growth rate and feed conversion efficiency | [97] |
| *Sargassum wightii* | Sargassaceae | | *Penaeus monodon* | Giant tiger prawn | 400 mg/L | - | Better growth rate and increased disease resistance | [98,99] |
| *Spirulina* sp. | Spirulinaceae | Whole | *Cyprinus carpio* | Common carp | 5 g/kg | - | Higher growth rate and feed conversion ratio | [100] |

**Table 2.** *Cont.*

| Algae | Algae (Family) | Mode of Algae in Aqua Feed | Aquatic Organisms (Scientific Name) | Aquatic Organisms (Common Name) | Dietary Inclusion Level (g/kg or %) | Replaced Ingredients | Beneficial Effects | References |
|---|---|---|---|---|---|---|---|---|
| *Scenedesmus obliquus* | Scenedesmaceae | Whole and LEA | *Anarhichas minor* | Spotted wolffish | 4% | PR | Increased growth performance and skin color | [101] |
| *Schizochytrium* sp. | Thraustochytriaceae | Whole | *Salmo salar* | Atlantic salmon | 100–150 g/kg | FO | Enhanced growth performance, fillet quality, nutrient retention efficiency, and blood chemistry | [102] |
| *Schizochytrium* sp. | Thraustochytriaceae | Whole | *Oncorhynchus mykiss* | Rainbow trout | NM | - | Better replacement of FM and FO | [103] |
| *S. platensis* | Microcoleaceae | Whole | *Pterophyllum scalare* | Angel fish | 5 g/kg | FM | Better nutrient additives for fish growth | [104] |
| *S. platensis* | Microcoleaceae | Whole and LEA | *O. niloticus* | Nile Tilapia | 5–10 g/kg | PR | Better growth and immunity promoter | [105] |
| *S. platensis* | Microcoleaceae | Whole | *Trachinotus ovatus* | Silver fish | 5% | FM | Improved growth, body composition and feed utilization | [106] |
| *S. platensis* | Microcoleaceae | Whole | *C. carpio* | Koi carp | 5% | FM | Stimulation of the immune system | [100] |
| *Tisochrysis lutea* | Isochrysidaceae | - | *Sparus aurata* | Gilthead seabream | 5% | - | Higher growth rate, nutrient utilization, and survival rate | [107] |
| *Ulva rigida* | Ulvaceae | - | *Mugil cephalus* | Common grey mullet | 10 mg/kg | - | Enhanced growth response, and antioxidant and immune stimulation | [108] |

Abbreviation: Penaeus monodon—P. monodon, Cyprinus carpio—C. carpio, Oreochromis niloticus—*O. niloticus*, Salmo salar—S. salar, Pseudodiaptomus hessei—P. hessei, gram—g, microgram—μg, milligram—mg, kilogram—kg, milliliter—ml, Immunoglobulin—Ig, interleukin—IL, fishmeal—FM, fish oil—FO, lipid-extracted microalgae—LEA, not mention—NM, larval food—LF, protein—PR, digestible protein—DP, apparent digestibility coefficients—ADC.

### 4.2. Treatment of Aquatic Animal Diseases

The administration of microalgae meal in a 70-day feeding trial with post larval stage of Pacific white shrimp (*Litopenaeus vannamei*) revealed significant positive impact on disease resistance against *V. harveyi* infection and various immune parameters, total hemocyte count, phenoloxidase activity, superoxide dismutase activity, and bactericidal activity [109–112]. Polysaccharides derived from the tropical brown algae *Turbinaria ornata* demonstrated potent antibacterial activity against *A. hydrophila*, *P. aeruginosa*, *Streptococcus* sp., *Yersinia enterocolitica*, *Enterobacter* sp., *Proteus* sp., *V. cholerae*, *V. parahaemolyticus*, *V. alginolyticus*, and *E. coli*. These findings displayed the potential of algae-derived compounds in combating bacterial infections and promoting the health of aquatic animals [111,113–115]. Various techniques are available for the extraction of polysaccharides from seaweeds, including hot water extraction, enzymatic-assisted extraction, microwave-assisted extraction, and ultrasonic-assisted extraction. Each of these methods has advantages and may be chosen based on the specific requirements of the polysaccharide extraction process [112,116]. *Scenedesmus quadricauda*, a green alga, is not only a good antioxidant but has also demonstrated bactericidal activity against bacteria like *Staphylococcus aureus* and *P. aeruginosa*. Additionally, the biomass of *S. quadricauda* is a rich source of protein, making it suitable for use as a feed supplement in aquatic animal's diets and is a valuable resource for enhancing the nutritional quality of the feeds and promoting health and growth of aquatic species [117]. The dietary incorporation of microalgae such as *Sargassum cinereum*, *Padina gymnospora*, and *Gracillaria folifera* has demonstrated significant antibacterial activity against *Pseudomonas* spp. in Mozambique tilapia (*Oreochromis mossambicus*), and these species are important due to their antibacterial and antiviral properties [118,119]. The dietary incorporation of *P. gymnospora* polysaccharide at various concentrations (0.01, 0.1, and 1.0%) showed that inclusion at 0.1% and 1.0% levels increased the production of antibodies and improved the resistance of common carp (*Cyprinus carpio*) against *A. hydrophila* and *E. tarda*. This highlights the potential of *P. gymnospora* polysaccharides as a beneficial dietary supplement by enhancing the immune response and disease resistance in aquatic species [120]. The dietary inclusion of various algae compounds, including polysaccharides, fatty acids, phlorotannins, pigments, lectins, alkaloids, terpenoids, and halogenated compounds, significantly enhanced the antioxidant capacity, immune response, survival rate, and infection resistance against pathogens in Mozambique tilapia [113,121]. Supplements of *Azolla pinnata* and *Nannochloropsis oculata*, used in the remediation of environmental pollutants are reported to improve antioxidant capacity, growth performance, immune system function, survival rates, and disease resistance in Nile tilapia [114]. The combination of dietary additives of *Chlorella* and *Spirulina* has demonstrated protective effects against *A. hydrophila* infection in Nile tilapia as these algae-based additives improved the immune response and disease resistance of the fish, contributing to their health and survival [122]. In contrast, dietary supplementation of *Dunaliella salina* at a dosage of 300 mg/kg for 8 weeks did not lead to changes in phenoloxidase activity and hemocyte count, but administration increased protection against *White Spot Syndrome Virus* (WSSV) in juvenile black tiger shrimp (*Penaeus monodon*) compared to control groups. This suggests that *D. salina* supplementation may not have direct effects on certain immune parameters but can still enhance the shrimp's resistance to specific pathogens like WSSV [9].

### 4.3. Growth Promoters

The dietary incorporation of *Spirulina* sp. at levels ranging from 5% to 15% in a 60-day feeding trial involving juvenile giant freshwater prawn (*Macrobrachium rosenbergii*) resulted in enhanced growth performance, reduced mortality, and significantly improved feed utilization. This indicates the potential of *Spirulina* sp. as a valuable dietary supplement for improving the health and overall production efficiency of the prawns in aquaculture [96]. *Spirulina* used as a feed supplement is known to be a valuable protein source, and in common carp, its inclusion in the diet has revealed improved growth and increased body weight by more than 25% vs. the control group. [93,123]. The dietary incorporation of *Spirulina* at levels ranging from 5% to 20% significantly enhanced the survival rate, growth,

and feed intake of juvenile Pacific white shrimp and revealed a positive impact on the health and production efficiency of shrimp in aquaculture [124]. In an 11-week study using juvenile Pacific white shrimp, the partial replacement of FM with *S. platensis* resulted in a significantly improved growth response. This suggests that incorporating *S. platensis* into the diet can be an effective strategy to enhance the growth performance of shrimp in aquaculture and reduce the dependency of FM [6,100]. The dietary incorporation of the microalgae *Scenedesmus almeriensis* at various combinations (ranging from 0% to 39%) in a 45-day experiment had a significant positive impact on the growth, body composition, gut microbiota composition, and production of short-chain fatty acids in gilthead sea bream (*Sparus aurata*) juveniles [125]. The single or combined dietary inclusion of *Isochrysis galbana*, *Pavlova lutheri*, and *Chaetoceros muelleri* significantly enhanced the growth performance and survival of grooved carpet shell (*Ruditapes decussatus*) larvae well as the biochemical and fatty acid composition during the larval development and underscore the value of algae-based dietary supplements in promoting the health and development of aquatic larvae in aquaculture [10]. The dietary incorporation of the macroalgae *Gracilaria arcuata* as feed supplement, 10, 20 and 30% to African catfish (*Clarias gariepinus*) revealed significantly lower growth, feed utilization, feed intake, and protein efficient ratio at 20 and 30% inclusion vs. control group and 10% inclusion [126]. In contrast, a later study, using *G. arcuata* at different levels (20, 40, and 60%) in a 12-week trial with Nile tilapia led to a 20% improvement in growth, enhanced body composition, and increased feed intake compared to control-fed fish [127,128]. In a 60-day feeding trial, the use of flocculated algae-enriched live feed *Artemia franciscana* resulted in improved growth, survival rates, maintenance of gut microbiota composition, increased villi length, enhanced digestive enzyme activities, improved feed conversion ratio, feed consumption ratio, and better nutrition indices compared to the control group and indicated that the use of algae-enriched live feed can have multiple beneficial effects on the growth, health, and overall performance of catla [129]. Dietary incorporation of different algae or single alga such as *A. pinnata*, *C. vulgaris*, *S. platensis*, and *Scenedesmus* spp., as well as their extracts, are revealed to be nutrient-rich feed sources for aquatic animals. These feed supplements can partially replace FO and FM, as they significantly improve antioxidant capacity, immune response, nutrient utilization, body weight, and growth performance of aquatic animals and highlight the potential of algae-based feed supplements by enhancing health and production of aquatic species [95,115,130]. Dietary inclusion of the microalgae *Pyropia yezoensis* extract at different combinations (ranging from 0 to 20 g/kg) in a 9-week feeding trial with olive flounder revealed that a 15% dietary inclusion notably enhanced weight gain, feed efficiency, and immunity and increased the level of PUFAs [92]. Dietary supplementation of red seaweed (*Gracilaria pygmaea*) at a level of 90 g/kg (GL-90) significantly improved the growth performance of rainbow trout fry [131]. However, the supplementation did not reveal a significant effect on antioxidant responses and digestive activities. Dietary inclusion of 3% *Arthrospira platensis* significantly improved the growth of Nile tilapia, but administration did not modulate the gut microbiota [132]. Dietary additives of microalgae *Spirulina* and *Schizochytrium*, used as replacements for FM, FO or plant oil, significantly improved fish growth and maintained fillet quality and displayed the potential of using these microalgae as sustainable alternatives in aquafeeds to enhance fish growth and the quality of fillets while reducing reliance on traditional protein sources like FM and FO [133]. Dietary supplementation of *Sargassum ilicifolium* (10%) during an eight-week feeding trial with great sturgeon revealed several positive effects, including enhanced growth performance, hematological parameters (red blood cells (RBCs), white blood cells (WBCs), and hemoglobin levels), immune responses (respiratory bursts, complementary activities, and immunoglobulin M, lysozyme, and serum total protein levels), and defense against *Y. ruckeri* and highlight the potential of *S. ilicifolium* as a valuable dietary supplement for improving the health and growth performance of great sturgeon [134]. Dietary inclusion of macroalgae *Gracilaria vermiculophylla* at a level of 5% in a 91-day feeding trial improved the growth performance of rainbow trout, enhanced the innate immune response, and resulted

in the highest peroxidase, complement, and lysozyme activity [135]. Dietary incorporation of fucoidan-rich algae *Sargassum wightii* extract at various concentrations (1, 2, 3, and 6%) in a 45-day trial experiment using sutchi catfish (*Pangasianodon hypophthalmus*) fingerlings showed that dietary supplementation at 2% notably reduced stress levels, increased the survival rate against *A. hydrophila*, and enhanced immune parameters such as lysozyme activity and respiratory burst activity [136,137]. In a study with Nile tilapia, brown algae *Padina pavonica* was administrated at 2, 4, 6, and 8 g/kg for 45 days, and growth was improved at 8 g/kg inclusion [138]. Furthermore, the study revealed higher values of hematological and biochemical parameters with the 8 g/kg inclusion diet.

### 4.4. Immunostimulants

Algae feed supplements in aquatic animals play an important role in enhancing the immune system and infection resistance against viral pathogens. These supplements contain various bioactive compounds and nutrients that can bolster health and promote the disease resistance of aquatic animals [139–141]. Dietary incorporation of algae-derived astaxanthin at a concentration of 80 mg/kg improved hemolymph immunological indices, including total hemocyte counts, phagocytic activity of hemocytes, serum anti-superoxide radical activity, serum phenoloxidase activity, serum antibacterial activity, and serum bacteriolytic activity and disease resistance against WSSV, in Pacific white shrimp during a 4-week trial [142], showing the immunomodulatory and disease resistance-enhancing properties of astaxanthin derived from algae. Dietary incorporation of 5% *S. platensis* and *Cladophora* in a 60-day experiment significantly improved the immune system response in various fish species, including Nile tilapia, rainbow trout, channel catfish, and great sturgeon [143–146]. This improvement was evidenced by changes in parameters such as RBC and WBC counts, serum bacterial activity, phagocytosis, and lysozyme activity. In addition to immune system enhancement, the dietary supplementation also led to improved growth performance and infection resistance in these fish species, highlighting the potential of algae-based dietary supplements in enhancing the overall health and disease resistance of various fish species. In an 8-week feeding trial, dietary incorporation of *Laurencia caspica* extract at a concentration of 1.5% significantly improved the immune system of rainbow trout by improving immune gene expression, including IL-1β, lysozyme II, Complement C3, and TNF-α [147]. Furthermore, the supplementation of *L. caspica* extracts improved resistance towards A. *hydrophila* infection. A recent study by Khanzadeh et al. [148] revealed significantly higher effects of *L. caspica* extract on immunoglobulin M and complement 3 as well as increased activity of alkaline phosphatase and alanine aminotransferase in Nile tilapia. Furthermore, extract administration considerably improved the survival of the fish challenged with *Streptococcus agalactia*. Dietary supplementation of sulphated polysaccharides derived from green algae, specifically *Codium fragile*, resulted in the upregulation of immune-related gene expression in olive flounder, increased expression of genes associated with the immune response, including TNF-α, IFN-γ, IL-1β, IL-8, and enhanced lysozyme activity [149,150]. The dietary supplementation of green microalgae *Chlorella vulgaris* (10%) in a Nile tilapia diet revealed several beneficial effects, particularly when the fish was exposed to arsenic, as the supplementation resulted in enhanced serum biomarkers such as aspartate aminotransferase (AST), alanine aminotransferase (ALT), alkaline phosphatase (ALP), creatinine, and urea [151]. Additionally, the immune gene expression was significantly upregulated, with TNF-α increasing 14-fold, TGF-β1 increasing 13-fold, and IL-1β increasing 7-fold, indicating the algae's ability to mitigate the negative impact of arsenic exposure in Nile tilapia [152]. Regarding the administration of *S. platensis*, a recent study [105] revealed that the supplementation in a Nile tilapia diet improved interleukin 10, lysozyme activity, complement 3, and IgM serum levels, as well as survival against *A. hydrophila* infection.

### 4.5. Rotifers and Algae

According to Gilbert [153], rotifers are subdivided into four categories as defined by types of food ingested: (1) nanoplankton, (2) microplankton, (3) 5–50 μm algae, and (4) 5–

250 μm algae. Larval rearing of marine fish, for example, turbot (*Scophthalmus maximus*) and Atlantic halibut (*Hippoglossus hippoglossus*), can result in a bottleneck, and at this early life stage, it is important to transfer essential nutrients from algae to live food [154]. The reference to Yan et al. [155] likely provides valuable insights into the specific methodologies, experimental setups, and findings related to using rotifers for toxicity testing of harmful algae. Authors interested in this topic can benefit from consulting these papers to gain a deeper understanding of the experimental protocols, data interpretation, and potential applications of rotifer-based toxicity testing in the context of harmful algal blooms.

## 5. Conclusions

Microalgae have garnered attention for their potential benefits in aquaculture, contributing to various aspects of fish health and overall industry sustainability (as a growth enhancer, immunostimulant, antioxidant, and source of disease resistance). Incorporating microalgae into aquaculture practices aligns with the broader goals of sustainable and responsible aquaculture, addressing both economic and environmental considerations [155–159]. Microalgae in aquaculture practices offer a holistic approach to water quality management, nutrient cycling, and environmental sustainability. These organisms contribute to creating a more balanced and self-regulating aquatic ecosystem within aquaculture systems [160]. Consequently, microalgae play a pivotal role in supporting the early life stages of aquatic organisms in aquaculture. Their nutritional content, suitability for larviculture, and contribution to sustainable feed production make them a valuable component in the cultivation of various species, ultimately contributing to the overall success and sustainability of aquaculture operations [133,161–188] and contributing to the conservation of marine biodiversity. Microalgae also help alleviate the pressure on wild fisheries, support lower trophic levels, and reduce the ecological footprint of aquaculture operations, all of which are essential for maintaining the health and diversity of marine ecosystems. Another benefit of microalgae via biotechnology is decrease in the production cost by producing biomass using wastewater under the phytoremediation process. Microalgae's phytoremediation capabilities offer a cost-effective and sustainable solution for managing aquaculture wastewater. Microalgae biomass offers a wide range of positive impacts on aquaculture, spanning from the improvement of aquatic animal health to enhancing the quality of aqua feeds and wastewater treatment. It is important to note that specific regulations can vary between region and country. Aquaculture operators and scientists should be aware of and comply with local regulations to promote the responsible and environmentally friendly use of algae in aquaculture. The use of algae and algae-derived components in aquaculture is indeed a promising avenue for the future, contributing to more sustainable and environmentally responsible practices while simultaneously improving aquatic feed quality and animal health. This aligns with the growing emphasis on responsible and ethical food production, essential for meeting the increasing global demand for seafood.

**Author Contributions:** S.V.: conceptualization, writing—original draft, literature search, table preparation. E.R.: writing—review and editing, manuscript outline, supervision. H.G.: writing—review and editing. S.H.H.: writing—review and editing, validation. S.A.: writing—review and editing. C.-C.C.: writing—review and editing, validation, supervision, manuscript outline. All authors have read and agreed to the published version of the manuscript.

**Funding:** This research received no external funding.

**Institutional Review Board Statement:** Not applicable.

**Data Availability Statement:** Data are contained within the article.

**Conflicts of Interest:** The authors declare no conflicts of interest.

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
