# Peer review of "Use of Algae in Aquaculture: A Review"

_fishes, doi:10.3390/fishes9020063_

Round 1

Reviewer 1 Report

Comments and Suggestions for Authors

Review-January 2024

The paper: “Use of algae in aquaculture”, written by authors representing an interesting review for use of algae in aquaculture for feeding and waste remediation. The strength of this paper is on the detail information about the potential of algae to be used in the aquaculture.

The main comments are related to some detailed information needs to be provided as the micro and macroalgae are not separated, however the biochemical composition of these two groups of species has different content and obviously the application.  Additionally, the application of algae is required the certain scale of cultivation, this aspect is not described. Complementary the cost of algal production and potentially seaweeds harvesting is needed to be added and discussed in the review to make the feasibility for the commercial exploitation of application of algal biomass for the aquaculture application.  Another aspect is the regulation of the use of algal biomass grown on the waste nutrients to be used as aquaculture feeding.   Could you please add this information.

More specific comments: 

Abstract:

Use of some worlds are needed to be adjusted for more scientific vocabulary:

Vital food production – could be changed or explained.

Algae are abundant source of nutrients, could you please change this, because it is not exactly scientifically correct.

Algae represent a fresh- algal biotechnology is developed since beginning of 19th century. The wider application is still in progress, because the process of production is still expensive, and some technologies needed to be adjusted.

1.      Introduction:

 ~global food supply chain- where it is the application?

~the utilization of algal technology – what exactly? Algal biomass? Could you please b more precise?

~diatoms cannot be the macroalgae is the group of brown macroalgae. Please change.

~Microalgae serve… please add the evolutionary role of algal species.

~therefore, you wrote algae species serve… it is not species, it is algal biomass -please change.

 In the last paragraph of introduction, could you please add the information about the regulatory aspects for the algae grown on the wastewater to be used for aquafeed application.

2.      Cultivation process.

More citations are required, e.g. Silkina et al, 2019 etc.

3.      Wastewater…

More explanations are needed to be added for the enhancement of water quality, especially for COD. Could you please add the scale application efficiency and commercial exploitation of this aspect of algal cultivation. And the regulatory aspects are needed to be added.

 For table 1. Could you please add the scale of cultivation for the finding of the species use.

4.      Applications of algae in aquaculture

% of algal biomass inclusion is required for the full understanding of the effect on fish health.  The reference for the study on yellow cichlid is needed to be added.  Dried

algae, please replace by biomass.  The study of Nile tilapia for the % of replacements needed to be added. Figure 1 is too complicated for the understanding need to be simplified or split in two.

Some more references please add.

For the part 4.3 growth promotes, Gracilraia arculta is not microalgae it is macroalgae, please change.

What it is catla? Please change

Gracilaria vertmiculophylla is also maroalgae, please change.

4.4.                        Immunostimulants

More details are needed for multiple benefits for European seabass.

5.      Conclusion

~Play a direct function-it is not clear, please change.

The aspects of regulation should be added. Additionally, the bottlenecks of the application of algal biotechnology for the aquaculture feeding need to be added.

Comments on the Quality of English Language

Review-January 2024

The paper: “Use of algae in aquaculture”, written by authors representing an interesting review for use of algae in aquaculture for feeding and waste remediation. The strength of this paper is on the detail information about the potential of algae to be used in the aquaculture.

The main comments are related to some detailed information needs to be provided as the micro and macroalgae are not separated, however the biochemical composition of these two groups of species has different content and obviously the application.  Additionally, the application of algae is required the certain scale of cultivation, this aspect is not described. Complementary the cost of algal production and potentially seaweeds harvesting is needed to be added and discussed in the review to make the feasibility for the commercial exploitation of application of algal biomass for the aquaculture application.  Another aspect is the regulation of the use of algal biomass grown on the waste nutrients to be used as aquaculture feeding.   Could you please add this information.

More specific comments: 

Abstract:

Use of some worlds are needed to be adjusted for more scientific vocabulary:

Vital food production – could be changed or explained.

Algae are abundant source of nutrients, could you please change this, because it is not exactly scientifically correct.

Algae represent a fresh- algal biotechnology is developed since beginning of 19th century. The wider application is still in progress, because the process of production is still expensive, and some technologies needed to be adjusted.

1.      Introduction:

 ~global food supply chain- where it is the application?

~the utilization of algal technology – what exactly? Algal biomass? Could you please b more precise?

~diatoms cannot be the macroalgae is the group of brown macroalgae. Please change.

~Microalgae serve… please add the evolutionary role of algal species.

~therefore, you wrote algae species serve… it is not species, it is algal biomass -please change.

 In the last paragraph of introduction, could you please add the information about the regulatory aspects for the algae grown on the wastewater to be used for aquafeed application.

2.      Cultivation process.

More citations are required, e.g. Silkina et al, 2019 etc.

3.      Wastewater…

More explanations are needed to be added for the enhancement of water quality, especially for COD. Could you please add the scale application efficiency and commercial exploitation of this aspect of algal cultivation. And the regulatory aspects are needed to be added.

 For table 1. Could you please add the scale of cultivation for the finding of the species use.

4.      Applications of algae in aquaculture

% of algal biomass inclusion is required for the full understanding of the effect on fish health.  The reference for the study on yellow cichlid is needed to be added.  Dried

algae, please replace by biomass.  The study of Nile tilapia for the % of replacements needed to be added. Figure 1 is too complicated for the understanding need to be simplified or split in two.

Some more references please add.

For the part 4.3 growth promotes, Gracilraia arculta is not microalgae it is macroalgae, please change.

What it is catla? Please change

Gracilaria vertmiculophylla is also maroalgae, please change.

4.4.                        Immunostimulants

More details are needed for multiple benefits for European seabass.

5.      Conclusion

~Play a direct function-it is not clear, please change.

The aspects of regulation should be added. Additionally, the bottlenecks of the application of algal biotechnology for the aquaculture feeding need to be added.

Reviewer 2 Report

Comments and Suggestions for Authors

The review titled “Use of algae in aquaculture: A review” needs major revision before consideration of publication.

1.      In the beginning of the abstract, the authors mentioned the aquatic environmental pollution, and correlated the environmental issues with the use of algae in aquaculture. The logic between environmental issues with the use of algae in aquaculture is not so straight and obvious. The use of algae in aquaculture firstly responds to the challenge of feed ingredient shortage in aquafeed industry. Please revise the logic.

2.      For the sentence “Incorporating algae as dietary supplements leads to significant enhancements in antioxidant and antimicrobial activities”, I also did not see a tight correlation between algae as dietary supplement and antioxidant activities. The authors should put their attention on the major issues related to the use of algae in aquaculture, namely, the nutrient composition characteristics.

3.      In the abstract, this review should also include the challenges in the use of algae in aquaculture, namely, the factors inhibiting the large-scale use of algae in aquaculture.

4.      There is no line number throughout the MS, making it difficult to mark comments.

5.      In the introduction part, too detailed information should be avoided, in order not to repeat the contents in the following sections. Instead, in this section, the authors need to summarize the product types, namely, algae meal, algae oil, liquid algae suspension, algae extract, etc., and compare their traits, functions, and inhibiting factors.

6.      For the section “2. Cultivation process”, the authors just described what every process is, which everyone in this research field knows, rather than the research progress in this field. The introduction of the definition of each process does not add valuable knowledge, while the summary of the research progress does.

7.      “Table 2 Overview of algae used as a dietary supplement in aquaculture” should contain more detailed information such as fish size, feeding duration, dietary protein and lipid level, etc.

8.      “4.1 Replacements of fish meal and fish oil”, this part should be categorized by algae type, namely, macroalgae (species……), microalgae (species……).

9.      The following parts “4.2. Treatment of aquatic animal diseases”, “4.3. Growth promoters”, and “4.4. Immunostimulants” are too general, delivering litter useful information. 

Reviewer 3 Report

Comments and Suggestions for Authors

I noticed quite a few grammatical errors. For example:

Abstract: "Currently, totally 40 genera of algae species..." should be "Currently, 40 genera of algae species..."; "enhance growth performance..." should be "enhanced growth performance..."

I like your Figure 1. Applications of algae in aquaculture.

Table 2: The 'active' links for Sargassaceae, Ulvaceae and other algae need to be removed.

Conclusions (p. 12): "Consequently, microalgae species are presently used for the culture of the fish larvae [no comma needed after larvae] and juvenile shells, in addition to farming zooplankton [what does 'farming' zooplankton mean? Reword or explain.] necessary for growth of juvenile animals."

Use of the Rotifer Brachionus plicatilus fed different species of algae as a start-food for fish such as Turbot (Piggvar) should be discussed.

Comments on the Quality of English Language

From above:

I noticed quite a few grammatical errors. For example:

Abstract: "Currently, totally 40 genera of algae species..." should be "Currently, 40 genera of algae species..."; "enhance growth performance..." should be "enhanced growth performance..."

I like your Figure 1. Applications of algae in aquaculture.

Table 2: The 'active' links for Sargassaceae, Ulvaceae and other algae need to be removed.

Conclusions (p. 12): "Consequently, microalgae species are presently used for the culture of the fish larvae [no comma needed after larvae] and juvenile shells, in addition to farming zooplankton [what does 'farming' zooplankton mean? Reword or explain.] necessary for growth of juvenile animals."

Round 2

Reviewer 2 Report

Comments and Suggestions for Authors

No further comments.

Author Response

Thank you for your comments.